Change the direction: 3D optimal control simulation by directly tracking marker and ground reaction force data

Nitschke Marlies 1 marlies.nitschke@fau.de
Marzilger Robert 2
Leyendecker Sigrid 3
Eskofier Bjoern M. 1
Koelewijn Anne D. 1
1 Machine Learning and Data Analytics Lab, Department of Artificial Intelligence in Biomedical Engineering (AIBE), Friedrich-Alexander-Universität Erlangen-Nürnberg (FAU) , Erlangen , Germany
2 Division Positioning and Networks, Fraunhofer IIS, Fraunhofer Institute for Integrated Circuits IIS , Nuremberg , Germany
3 Institute of Applied Dynamics, Department of Mechanical Engineering, Friedrich-Alexander-Universität Erlangen-Nürnberg (FAU) , Erlangen , Germany
Climstein Mike
Electronic publication date: 2023 Feb 7
Publication date: 2023
Volume: 11
Electronic Location ID: e14852
Received 2022 Aug 1; Accepted 2023 Jan 13
Copyright: © 2023 Nitschke et al.
Copyright year: 2023
Copyright holder: Nitschke et al.
License: This is an open access article distributed under the terms of the Creative Commons Attribution License, which permits unrestricted use, distribution, reproduction and adaptation in any medium and for any purpose provided that it is properly attributed. For attribution, the original author(s), title, publication source (PeerJ) and either DOI or URL of the article must be cited.
License URL: https://creativecommons.org/licenses/by/4.0/

Keywords: Dynamic optimization, Trajectory optimization, Human Movement Simulation, Motion capturing, Musculoskeletal model, Locomotion, Three-dimensional

Funding: Center for Analytics–Data–Applications (ADA-Center) German Research Foundation (DFG, Deutsche Forschungsgemeinschaft) SFB 1483–Project-ID 442419336 Heisenberg professorship programme ES 434/8-1 adidas AG Institute of Applied Dynamics financially INST 90/985-1 FUGG Friedrich-Alexander-Universität Erlangen-Nürnberg (FAU) Marlies Nitschke and Robert Marzilger were supported by the Bavarian Ministry of Economic Affairs, Regional Development and Energy through the Center for Analytics—Data—Applications (ADA-Center) within the framework of “BAYERN DIGITAL II”. Sigrid Leyendecker, Bjoern M Eskofier, and Anne D Koelewijn were supported by the German Research Foundation (DFG, Deutsche Forschungsgemeinschaft) under Grant SFB 1483–Project-ID 442419336. Bjoern M Eskofier was also supported by the German Research Foundation (DFG) within the framework of the Heisenberg professorship programme (grant number ES 434/8-1). Anne D Koelewijn was also supported by faculty endowment from adidas AG. We carried out all measurements in the motion analysis laboratory at the Institute of Applied Dynamics financially supported by the German Research foundation INST 90/985-1 FUGG. Furthermore, the HPC resources were provided by the Erlangen National High Performance Computing Center (NHR@FAU) of the Friedrich-Alexander-Universität Erlangen-Nürnberg (FAU). The funders had no role in study design, data collection and analysis, decision to publish, or preparation of the manuscript.

==============================
Optimal control simulations of musculoskeletal models can be used to reconstruct motions measured with optical motion capture to estimate joint and muscle kinematics and kinetics. These simulations are mutually and dynamically consistent, in contrast to traditional inverse methods. Commonly, optimal control simulations are generated by tracking generalized coordinates in combination with ground reaction forces. The generalized coordinates are estimated from marker positions using, for example, inverse kinematics. Hence, inaccuracies in the estimated coordinates are tracked in the simulation. We developed an approach to reconstruct arbitrary motions, such as change of direction motions, using optimal control simulations of 3D full-body musculoskeletal models by directly tracking marker and ground reaction force data. For evaluation, we recorded three trials each of straight running, curved running, and a v-cut for 10 participants. We reconstructed the recordings with marker tracking simulations, coordinate tracking simulations, and inverse kinematics and dynamics. First, we analyzed the convergence of the simulations and found that the wall time increased three to four times when using marker tracking compared to coordinate tracking. Then, we compared the marker trajectories, ground reaction forces, pelvis translations, joint angles, and joint moments between the three reconstruction methods. Root mean squared deviations between measured and estimated marker positions were smallest for inverse kinematics (e.g., 7.6 ± 5.1 mm for v-cut). However, measurement noise and soft tissue artifacts are likely also tracked in inverse kinematics, meaning that this approach does not reflect a gold standard. Marker tracking simulations resulted in slightly higher root mean squared marker deviations (e.g., 9.5 ± 6.2 mm for v-cut) than inverse kinematics. In contrast, coordinate tracking resulted in deviations that were nearly twice as high (e.g., 16.8 ± 10.5 mm for v-cut). Joint angles from coordinate tracking followed the estimated joint angles from inverse kinematics more closely than marker tracking (e.g., root mean squared deviation of 1.4 ± 1.8 deg vs. 3.5 ± 4.0 deg for v-cut). However, we did not have a gold standard measurement of the joint angles, so it is unknown if this larger deviation means the solution is less accurate. In conclusion, we showed that optimal control simulations of change of direction running motions can be created by tracking marker and ground reaction force data. Marker tracking considerably improved marker accuracy compared to coordinate tracking. Therefore, we recommend reconstructing movements by directly tracking marker data in the optimal control simulation when precise marker tracking is required.

Introduction

Kinematics and kinetics of walking or running are estimated from measurements with optical motion capture in various fields of biomechanical research. While research often focuses on straight walking or running, change of direction (COD) motions are also crucial in everyday life. Cutting maneuvers are, for example, performed frequently in multi-directional team sports (Fox, 2018). Non-contact COD maneuvers or rapid decelerations have been identified as the primary cause of anterior cruciate ligament (ACL) injuries (Donnelly et al., 2017; McLean et al., 2004). In the last years, a great number of biomechanical studies (e.g., Barengo et al. (2014)) were conducted to develop and analyze the effect of injury prevention training programs like FIFA 11+ (Bizzini, Junge & Dvorak, 2011). However, in those studies, little attention was spent on the method that was used to estimate the kinematic and kinetic variables like joint angles, joint moments, or muscle forces.

Inverse methods, i.e., inverse kinematics and dynamics, combined with static or dynamic optimization using a human model are widely used to obtain joint and muscle kinematics and kinetics from marker positions and ground reaction forces (GRFs) (Seth et al. (2018); see Fig. 1A). However, whereas these methods have the advantage that they are rapid to solve and easy to apply, they also have major weaknesses. Inverse kinematics estimates the generalized coordinates of the model, i.e., global translation, global orientation, and joint angles, for each time step separately. Since time dependency is not taken into account, inverse kinematics is prone to track measurement noise. This means that, when measurement noise causes a sudden change in a marker position, inverse kinematics will also estimate a sudden change in the respective joint angle, even though it is unrealistic for humans to move in a non-smooth fashion. In a second step, joint moments are estimated with inverse dynamics. While inverse dynamics takes time dependency into account, it allows for dynamic inconsistencies, i.e., inconsistencies between kinematics and kinetics. These inconsistencies are caused by modeling errors and inaccuracies in the measured data and typically result in residual forces and moments at the last segment (Faber, van Soest & Kistemaker, 2018). However, there is no physical cause for these residuals and it is difficult to trace which model parameter or measurement error contributed to the residuals. Therefore, residuals are hard to interpret and researchers are advised to prevent residuals that are large enough to influence the study conclusion (Hicks et al., 2015). After inverse dynamics, muscle forces can be computed using static or dynamic optimization to resolve the muscle redundancy problem. Past studies found minor differences in muscle forces obtained from dynamic compared to static optimization for walking (Anderson & Pandy, 2001) and even for running (Lin et al., 2012). However, more recent evidence highlights that muscle activation, and thus muscle efficiency, is influenced when neglecting tendon compliance (De Groote et al., 2016; Miller, Umberger & Caldwell, 2012). Hence, modeling muscle dynamics is especially important for faster motions such as running or sprinting.

Figure 1 Processing pipelines of the three reconstruction methods.

In this article, we compare inverse methods (A) with coordinate tracking simulation (B), and marker tracking simulation (C) to reconstruct measurements of straight running, curved running and a v-cut.

Open-loop optimal control simulation of a human model is an alternative to inverse methods for estimating various biomechanical variables from a measured motion, which results in mutually and dynamically consistent kinematics and kinetics. In open-loop optimal control simulations, also called trajectory optimization, joint and muscle kinematics and kinetics of the model are obtained by minimizing an objective while accounting for system dynamics. When open-loop optimal control simulations are used to reconstruct movements, it is assumed that the skeletal or musculoskeletal model is perfect and does not contain modeling errors. For reconstructing a measured movement with optimal control simulations, the objective often combines a tracking term minimizing the difference between measured and simulated data with an energy-related term. Optimal control simulations have gained increased attention in recent years due to methodological advances. The exploration of direct collocation methods combined with the implicit formulation of the system dynamics allows the optimal control problem to be solved efficiently (De Groote et al., 2016; Nitschke et al., 2020; van den Bogert, Blana & Heinrich, 2011). Furthermore, the increasing availability of toolboxes facilitates access to the methodology (Dembia et al., 2020; Michaud et al., 2022; Patterson & Rao, 2014). Optimal control simulations cannot only be used to reconstruct measured motions but also to predict responses to environmental changes (Dorschky et al., 2019; van den Bogert, Blana & Heinrich, 2011; van den Bogert et al., 2012) or task changes (Lin, Walter & Pandy, 2018; Nitschke et al., 2020). Predictive simulations can either track related data as reference or predict novel movements without any input from measurements.

Reconstructive optimal control simulations are traditionally generated by tracking generalized coordinates or joint angles in combination with measured GRFs (e.g., Dembia et al. (2020); Haralabidis et al. (2021); Heinrich, van den Bogert & Nachbauer (2014); Lin, Walter & Pandy (2018); Nitschke et al. (2020); van den Bogert, Blana & Heinrich (2011); van den Bogert et al. (2012); see Fig. 1B). Since generalized coordinates are used as kinematic states of the model and are thus optimization variables, tracking of coordinates is computationally more efficient than tracking of other biomechanical variables which are not part of the optimization variables. However, the coordinates have to be estimated from marker data before simulation using, for example, inverse kinematics. Hence, the inaccuracies in the estimated coordinates are tracked in the simulation resulting in error propagation. Additionally, individual joint angles are tracked rather than absolute positions. Therefore, tracking errors of each joint angle accumulate down the kinematic chain, which can cause larger positional differences at the end of this chain. In contrast to tracking coordinates in the simulation, tracking marker positions directly could avoid error propagation and error accumulation along the kinematic chain (see Fig. 1C). Recently, marker tracking was successfully investigated for upper limb models with up to 7 degrees of freedom (DoFs) and up to 20 muscle tendon units (MTUs) partly in combination with electromyography (EMG) tracking (Bailly et al., 2021; Bélaise et al., 2018a, 2018b; Hoffmann et al., 2020). Furthermore, marker tracking was investigated for a single leg model with 6 DoFs and 17 MTUs (Moissenet et al., 2019). Febrer-Nafría et al. (2020) and Venne et al. (2022) compared coordinate and marker tracking using full-body skeletal models for walking and somersaults, respectively. Their research indicated that marker tracking simulations followed measured marker positions more closely than coordinate tracking. Therefore, marker tracking was more accurate in terms of marker errors.

Overall, previous research on marker tracking was limited to small models with few optimization variables and was limited to an evaluation with simulated or little data. The models previously used for marker tracking were either musculoskeletal models with only a few DoFs (Bailly et al., 2021; Bélaise et al., 2018a, 2018b; Moissenet et al., 2019) or skeletal models (Febrer-Nafría et al., 2020; Hoffmann et al., 2020; Venne et al., 2022). Therefore, it is unclear whether optimal control simulation with marker tracking is numerically feasible for full-body musculoskeletal models, resulting in a considerably larger number of optimization variables and constraints. However, entire 3D body kinematics and kinetics should be considered especially for an accurate analysis of COD running motions since upper-body kinematics can influence, for example, knee moments (Donnelly et al., 2012). Moreover, COD running motions have not yet been reconstructed with optimal control simulation, which would particularly be relevant for sports science. Furthermore, evaluation was performed either with simulated data (Bailly et al., 2021; Bélaise et al., 2018a) or data of only one participant (Bélaise et al., 2018b; Febrer-Nafría et al., 2020; Hoffmann et al., 2020; Moissenet et al., 2019) except for Venne et al. (2022), who reconstructed in total 26 somersault trials of five participants. Consequently, there is no clear evidence of whether marker tracking is superior to coordinate tracking for running and especially for COD running motions.

We investigated the feasibility of directly driving 3D optimal control simulations by marker and GRF data using a full-body musculoskeletal model, especially for reconstructing COD running motions. We developed a method for creating optimal control simulations for arbitrary motions like COD motions with direct collocation and an implicit formulation of the system dynamics. Since gold standard measurements of kinematics are hardly available and joint kinetics cannot be measured directly, we compared marker tracking simulations to coordinate tracking simulations and inverse methods for estimated marker positions, GRFs, pelvis translation, angles, and joint moments (see Fig. 1). To create strong evidence with our study, we performed the analysis for 10 participants and three trials each of straight running, curved running, and a v-cut.

Methods

In this section, we first describe the experimental data and the musculoskeletal model used for motion reconstruction. Then, we give details about the inverse methods, the optimal control simulations, and the evaluation.

Experimental data

We recorded motion capture data of 10 healthy young participants (four female, six male; age: 27.5 ± 3.5 years; height: 1.76 ± 0.10 m; mass: 71.3 ± 12.1 kg). The ethics committee of the Friedrich-Alexander-Universität Erlangen-Nürnberg (Re.-No. 106_13 B) approved the study, and participants gave informed written consent before participation. We obtained marker positions of 42 reflective markers at 175 Hz with 11 infrared cameras (Qualisys, Gothenburg, Sweden) and GRFs of the right and left foot at 1750 Hz with two force plates (Bertec Corporation, Columbus, USA). Simultaneously, we recorded data from 11 inertial measurement units but did not use the data in this article.

Each participant first performed a static trial in a neutral pose (N-pose) with one foot on each force plate and the arms beside the body. Afterwards, the participants completed multiple trials respectively for straight running, curved running with a radius of 7 m, and a 90° v-cut (see Fig. 2). For curved running and the v-cut, we indicated the path with crepe tape on the floor.

Figure 2 Motion paths of the three motion types straight running (A), curved running (B), and v-cut (C).

The paths are highlighted in red. The blue boxes indicate the force plates. The scale of the illustration is given by the gray boxes which are one by one meter.

For every participant and motion type, we choose the three trials for which the force plates were hit entirely and marker occlusions were smallest. We filled gaps in the marker data using the Qualisys Track Manager but did not apply any filter. We defined, but did not extract, the motion of interest before reconstruction to reduce edge effects from filtering for inverse dynamics or from the initial constraint used in the simulation. The motions of interest started at the initial contact of the right foot at the first force plate and ended with the next initial contact of the right foot. For the v-cut, this corresponded to the execution and departure contact. We determined each initial contact at the time where the vertical velocity of the mean of the heel and toe marker position was at a minimum (O’Connor et al., 2007).

Musculoskeletal model

We used our 3D full-body musculoskeletal model called runMaD, which is short for “running model for motions in all directions” (Nitschke et al., 2020). In contrast to other musculoskeletal models, runMaD has an adapted pelvis rotation sequence that makes the pelvis obliquity and tilt interpretable according to their clinical definition independent of the movement direction, i.e., independent of the rotation around the vertical axis (Baker, 2001). The model has 33 DoFs (six DoFs between ground and pelvis, three DoFs at the lumbar joint, seven DoFs per leg, five DoFs per arm), 92 MTUs in the lower body (six MTUs at the lumbar joint, 43 MTUs per leg), and five torque actuators per arm. The muscle paths are described in OpenSim using point sets. For the optimal control simulation, muscle-tendon lengths are defined with polynomial functions depending on the joint angles, and a penetration-based ground contact model with eight contact points at each foot is used (see supplemental information of Nitschke et al., 2020). We scaled the model for every participant using the static trial in N-pose in OpenSim 4.3 (Seth et al., 2018) but did not personalize any muscle properties.

Inverse methods

Using the scaled models, we performed inverse kinematics and dynamics as reference with OpenSim 4.3 (Seth et al., 2018). Additionally, we used the resulting generalized coordinates as input for the coordinate tracking simulations (see Fig. 1). We weighted all markers equally in inverse kinematics. For inverse dynamics, we filtered the generalized coordinates and GRFs with a 3rd order dual-pass low-pass Butterworth filter with a cut-off frequency of 15 Hz (Derrick et al., 2020). We reconstructed the entire trials and not only the extracted motions of interest since the Butterworth filter has an infinite impulse response causing undesired effects, especially at the edges of the trajectories.

Optimal control simulations

We created coordinate and marker tracking simulations by solving optimal control problems with the scaled musculoskeletal models. We formulated the optimal control problem as a constrained non-linear optimization problem using direct collocation and a backward Euler discretization. A state trajectory x and a control trajectory u of the model are found by minimizing a multi-objective function J(x,u) with respect to the model dynamics f.

Objective Function The objective J was a weighted sum of tracking Jtra, muscular effort Jmus, torque effort Jtor, and regularization Jreg:

(1) J(x,u)=Jtra+Jmus+Jtor+Jreg,

(2) Jtra=∑j∈StraWtra,jNNtra,j∑k=1N∑i=1Ntra,j(yi,j[k]−y^i,j[k])2

(3) Jmus=WmusNNmus ∑k=1N ∑i=1Nmuswmus,i∑j=1Nmuswmus,j(ne,i[k])3,

(4) Jtor=WtorNNtor∑k=1N∑i=1Ntor(mi[k])2,

(5) Jreg=Wreg(N−1)T2(Nstates+Ncontrols) ∑k=1N−1(∑i=1Nstates(xi[k+1]−xi[k])2+∑i=1Ncontrols(ui[k+1]−ui[k])2).

The tracking term Jtra consisted of separate terms with individual weights Wtra,j for each data type j of the set Stra. Depending on the tracking method, we used the following data types (see Fig. 1): Coordinate tracking: 3D global translation of the pelvis, global orientation of the pelvis, and joint angles obtained from inverse kinematics and measured GRFs of right and left foot (i.e., Stra={translation,angle,GRF}, Ntra,translation=3, Ntra,angle=30, and Ntra,GRF=6).

Marker tracking: measured 3D marker positions of all 42 markers and GRFs of right and left foot (i.e., Stra={marker,GRF}, Ntra,marker=126, and Ntra,GRF=6).

We minimized the squared difference between reference signal y and estimated signal y^ in the tracking term for N collocation nodes and Ntra,j signals. Using Jmus with the respective weight Wmus, we resolved the muscle redundancy problem by minimizing the sum of the volume-weighted cubed neural excitations ne of each of the Nmus muscles. We used the muscle volume wmus,i of a muscle i to account in the effort term for the strongly varying sizes and maximum isometric forces of the MTUs and, therefore, spread muscle recruitment more evenly (Happee & Van der Helm, 1995).

Furthermore, we minimized the sum of the squared torque controls m actuating the Ntor arms in Jtor with the weight Wtor. The regularization term Jreg with the weight Wreg represents the minimization of the temporal derivative of the state and control trajectories, where Nstates and Ncontrols represent the number of states and controls, respectively. Such regularizations are used to improve the convergence of the optimization algorithm. The duration T of the motion was prescribed by the duration of the tracking data.

Model Dynamics We formulated the model dynamics implicitly and used backward Euler discretization. Hence, the following constraint was applied for each node:

(6) f(x[k+1],x[k+1]−x[k]h,u[k+1])=0∀k=1,...,N−1,

where h=T/(N−1). More details about the system dynamics and the implementation are given by Nitschke et al. (2020).

However, Eq. (6) does not contain the control u[1], as the following dynamics apply at the first node k=1:

(7) f(x[2],x[2]−x[1]h,u[2])=0.

Furthermore, the state x[1] at the first node appears only in one equation (equation k=1), while all other states x[k] appear in two subsequent equations (equation k−1 and equation k). Hence, additional information is required to ensure that the optimal control problem can be solved, which is commonly done in two ways. The first option is to add an initial state. The second option is to apply an additional constraint that describes the task. For example, gait is typically constrained to be periodic (e.g., van den Bogert, Blana & Heinrich (2011)). The usage of task constraints can especially be beneficial when a new motion should be predicted based on data of a related motion (Nitschke et al., 2020) or when a specific gait speed should be prescribed for standardization (Dorschky et al., 2019). In this work, however, we aimed to reconstruct arbitrary motions, which would not be possible if an initial state or task constraint had to be prescribed. Therefore, we did not use a task constraint but additionally ensured model dynamics using forward Euler discretization at k=1:

(8) f(x[1],x[2]−x[1]h,u[1])=0,

except for the identities q˙−dqdt=0 of the global pelvis translation and orientation to not restrict global motion. The combination of Eqs. (6) and (8) implies constant velocities of the states and controls between nodes 1 and 2. In contrast to prescribing specific values as initial states, this constraint does not require prior knowledge of the motion. However, the assumption of a constant velocity slightly influences the result at the first nodes. To avoid impact on the motion of interest, we included additional samples at the beginning of the signal.

Initialization We first simulated static standing for each participant and each tracking method to calibrate the ground contact model and to generate an initial guess for the running simulations. In the objective, we tracked the kinematic and GRF data of one time point of the N-pose. Since only one time point was simulated, we omitted the regularization term Jreg and ensured static equilibrium by f(x[1],0,u[1])=0, which ensures that the velocities and accelerations are zero. We adapted the ground contact model for different shoe sole thicknesses of the participants by optimizing a vertical offset for the position of the ground contact points during the simulation. The position of the ground contact points had to be adapted for coordinate and marker tracking since we tracked absolute positions of the pelvis or markers, respectively. We solved 10 simulations for each optimization problem using different random initial guesses and selected the solution with the lowest objective to reduce the chance of obtaining a local minimum.

Running Simulations We then generated simulations for three trials each of straight running, curved running, and the v-cut for each of the 10 participants and tracking method. In total, this resulted in 90 simulations each for marker and coordinate tracking. In the objective, we tracked reference kinematics and GRFs using the sampling frequency of 175 Hz and therefore downsampled the GRF data. The reference data was not filtered prior to the simulation since the simulation itself acts as a physical filter that takes the model dynamics into account. We found in pilot simulations that 10 additional samples before the motion of interest, i.e., before the initial contact, are sufficient to not cause observable artifacts in the motion of interest when using Eq. (8). The original sampling frequency combined with the additional samples resulted in optimization problems with 127 to 180 collocation nodes N, which corresponds to durations T of approximately 0.72 s to 1.02 s.

Solution Process We selected the weights W of the objective terms (see Eqs. (2)–(5)) empirically using data of the first participant such that the tracking data was followed while the neural excitation remained smooth (see Table 1). We used an equal weight for all tracking variables of the same type, meaning that we used one weight for pelvis translations, one for angles, one for markers, and one for GRFs. The constrained non-linear optimization problems were solved using IPOPT 3.12.3 (Wächter & Biegler, 2006) with a convergence tolerance for the scaled nonlinear program (NLP) error of 10−4 and a maximum number of iterations of 2⋅104. We used a high-performance cluster to parallelize the 180 running simulations. Each simulation was performed on a single cluster node with one Xeon E3-1240 CPU with four cores.

Table 1 Weights W of the multi-objective function (see Eqs. (2)–(5)).

We determined the weights empirically using the data of the first participant only.

	Standing	Running	
Coordinate	Marker	Coordinate	Marker	
Wtra,translation in mm −2	10−3	–	10−3	–	
Wtra,angle in deg −2	10−1	–	10−1	–	
Wtra,marker in mm −2	–	10−2	–	10−2	
Wtra,GRF in (BW%) −2	10−2	10−2	10−3	10−3	
Wmus	1	1	1	1	
Wtor	10−1	10−1	10−1	10−1	
Wreg	–	–	10−3	10−3	

Evaluation

We analyzed the convergence of coordinate and marker tracking problems by comparing the number of iterations and the wall and CPU time required to solve the running simulations. Whereas the wall time represents the time passed to solve the problem, the CPU time captures the total time the single cores are active, i.e., the total time required for calculations. The CPU time can therefore be greater than the wall time if multiple cores of a CPU are used for processing. Furthermore, we computed the CPU time per iteration to analyze the computational demand of a single iteration. To evaluate the computational demand of the objective, constraints, and their derivatives compared to the time spent in the optimization algorithm of IPOPT, we obtained the ratio of CPU time spent in the function evaluations of the NLP from the log file of IPOPT.

To investigate the estimated kinematics and kinetics, we extracted the motions of interest from the reconstructed motions. We visually compared the reconstructed motions of interest of inverse kinematics, coordinate tracking, and marker tracking using the visualization of the kinematics in OpenSim and trajectory graphs of marker positions, GRFs, pelvis translation, angles, and joint moments. Additionally, we obtained the root mean squared deviation (RMSD) for each motion of interest to analyze the agreement of the trajectories. We used the measured data as reference for marker positions and GRFs. For the generalized coordinates and joint moments, we considered the result of the inverse methods as reference since no ground truth was available. GRFs were scaled to body-weight percent (BW%) and joint moments were scaled to body-weight body-height percent (BW BH%). We aggregated all results by computing the mean and standard deviation over all variables of one type (e.g., marker positions) and all trials of one motion (e.g., straight running). Consequently, each mean value resulted from 30 simulations.

Finally, we evaluated residual forces and moments that are caused by dynamic inconsistencies. For inverse methods, we computed the root mean squared (RMS) residual forces and moments at the pelvis. In the coordinate and marker tracking simulations, we constrained the dynamic residuals to be zero by using the multibody dynamics as constraints in the optimization (see equation S1 in the supplemental information of Nitschke et al. (2020)). Nevertheless, the optimization result could slightly violate the multibody dynamics within the constraint violation tolerance of 0.001. Therefore, we analyzed the RMS residual pelvis forces, pelvis moments, and joint moments resulting from the constraint violations of the multibody dynamics. We scaled the residual forces to percent of maximal net ground reaction forces (GRF max%) and the residual moments to percent of maximal net ground reaction forces and body-height percent (GRF max BH%) based on Hicks et al. (2015).

Results

All 180 running simulations converged. The averages of the scaled NLP errors were between 7.3⋅10−5 and 8.0⋅10−5 for straight running, curved running, and v-cut and for marker and coordinate tracking (see Table 2). The NLP errors did not differ largely depending on the motion type or tracking method. Solving the marker tracking simulations required considerably more iterations and time than the coordinate tracking simulations. The fastest simulations for marker and coordinate tracking converged after a wall time of 2 h 42 min and 51 min, respectively, while the slowest simulations required 9 h 31 min and 3 h 56 min, respectively. The CPU time spent in every iteration was comparable between both tracking methods. However, the ratio of CPU time which was spent in the function evaluation of the NLP was higher for the marker tracking (10.1% to 10.5%) compared to the coordinate tracking (7.8% to 8.1%).

Table 2 Mean ± standard deviation of the convergence criteria.

The average was computed over all trials of the respective motions straight running (SR), curved running (CR), and v-cut (VC).

	Motion	Marker tracking	Coordinate tracking	
Scaled NLP error	SR	7.7 ± 1.5 ⋅10−5	7.5 ± 1.7 ⋅10−5	
	CR	7.3 ± 2.0 ⋅10−5	8.0 ± 1.6 ⋅10−5	
	VC	7.3 ± 2.3 ⋅10−5	7.7 ± 1.5 ⋅10−5	
Number of iterations	SR	7437 ± 2217	2204 ± 664	
	CR	8068 ± 2030	2054 ± 505	
	VC	5788 ± 1699	2370 ± 821	
Wall time in hh:mm:ss	SR	05:51:30 ± 01:48:54	01:45:42 ± 00:41:28	
	CR	06:21:01 ± 01:27:17	01:34:10 ± 00:25:10	
	VC	04:57:36 ± 01:25:08	02:00:25 ± 00:39:40	
CPU time in hh:mm:ss	SR	12:33:17 ± 03:47:35	03:54:00 ± 01:31:40	
	CR	13:40:55 ± 03:02:47	03:30:17 ± 00:56:32	
	VC	10:40:31 ± 02:57:06	04:25:22 ± 01:27:08	
CPU time per iteration in ss:fff	SR	06:115 ± 01:004	06:279 ± 00:763	
	CR	06:196 ± 00:746	06:145 ± 00:802	
	VC	06:725 ± 01:027	06:794 ± 01:014	
CPU time in NLP in %	SR	10.5 ± 1.6	7.8 ± 0.8	
	CR	10.4 ± 1.2	8.1 ± 1.0	
	VC	10.1 ± 1.4	7.9 ± 1.1	

The visual inspection of the reconstruction showed that the methods generally led to natural running motions. We overlaid the animated skeletons for inverse methods, coordinate tracking, and marker tracking for a more detailed kinematic analysis. Figure 3A shows an exemplary v-cut (participant 02, trial 130 in Nitschke et al. (2022)), also provided as a video in the supplemental information. The three reconstruction methods showed good agreement. Nevertheless, it could be observed that the result of the coordinate tracking deviated from that of inverse methods and marker tracking while the skeletons from inverse methods and marker tracking superimposed better. This can for example be seen for the right foot in the screenshots of the samples 31 and 121 in Fig. 3A.

Figure 3 Kinematics including horizontal offset for visualization (A) and a selection of trajectories of the right leg (B) for a v-cut.

The result of inverse kinematics, coordinate tracking, and marker tracking is represented in red, orange, and green, respectively. Measured marker positions are displayed in blue. The motion had in total 149 samples at 175 Hz. GRFs were scaled to body-weight percent (BW%) and joint moments were scaled to body-weight body-height percent (BW BH%).

For the different reconstruction methods, trajectories of marker positions, GRFs, joint angles, and joint moments showed similar patterns, but there were also substantial differences (see Fig. 3B). Mean RMSDs were in the same order of magnitude, except for marker positions (see Table 3). The mean RMSDs between estimated and measured marker positions were smallest for inverse methods (e.g., 7.6 ± 5.1 mm for v-cut) and a bit higher for the marker tracking simulation (e.g., 9.5 ± 6.2 mm for v-cut). However, coordinate tracking resulted in nearly twice as high RMSDs (e.g., 16.8 ± 10.5 mm for v-cut), which is also observable in the trajectories (see Fig. 3B). Foot markers, such as the ToeL, which is placed at the head of the fifth metatarsal, were tracked more closely by the inverse methods than by the tracking simulations. This can for example be seen for the right foot in the screenshot of sample 31 in Fig. 3A or in the trajectories of ToeL in Fig. 3B. For the GRFs, marker tracking showed slightly smaller RMSDs compared to coordinate tracking (e.g., 3.9 ± 2.6 BW% vs. 5.8 ± 2.6 BW% for v-cut). While there was no clear trend for the translation, coordinate tracking was generally closer to the reference angles, which were obtained with inverse kinematics, than marker tracking (e.g., RMSDs of 1.4 ± 1.8 deg vs. 3.5 ± 4.0 deg for v-cut). Especially the joint angles of the metatarsophalangeal (mtp) joint deviated more from the reference for marker tracking but also for coordinate tracking (see Fig. 3B). Inverse methods resulted in higher plantarflexion of the mtp joint compared to the tracking simulations, and coordinate tracking resulted in very high dorsiflexion during push-off. Furthermore, joint angles reconstructed with both tracking simulations were smoother than those obtained from inverse methods (see Fig. 3B). In principle, joint moments followed the same course for all reconstruction methods but showed different oscillations. The mean RMSDs of the joint moments were similar for the two tracking simulations but marginally smaller for marker tracking (e.g., 0.7 ± 0.7 BW BH% vs. 1.0 ± 1.1 BW BH% for v-cut). For both methods, arm joints showed smaller RMSDs for the moments than leg joints. The v-cut generally led to higher mean RMSDs compared to straight and curved running (see Table 3).

Table 3 Mean ± standard deviation of the root mean squared deviation (RMSD) between estimated and reference variable.

The average was computed over all variables of the specific type (e.g., over all marker positions) and all trials of the respective motions straight running (SR), curved running (CR), and v-cut (VC). Input variables used in the particular reconstruction methods are highlighted in gray.

	Motion	Inverse methods	Marker tracking	Coordinate tracking	Reference	
Marker in mm	SR	6.7 ± 4.7	7.6 ± 4.8	13.4 ± 8.8	Measured data	
	CR	6.8 ± 4.7	8.2 ± 5.5	13.5 ± 8.6	
	VC	7.6 ± 5.1	9.5 ± 6.2	16.8 ± 10.5	
GRF in BW%	SR		3.4 ± 1.8	4.4 ± 2.4	
	CR		4.2 ± 3.4	4.5 ± 2.3		
	VC		3.9 ± 2.6	5.8 ± 2.6		
Translation in mm	SR		2.6 ± 1.1	2.7 ± 1.3	Inverse methods	
	CR		3.0 ± 1.7	2.9 ± 1.6	
	VC		4.5 ± 1.9	3.2 ± 1.2	
Angle in deg	SR		2.3 ± 2.9	0.9 ± 0.7	
	CR		2.5 ± 3.0	0.9 ± 0.7	
	VC		3.5 ± 4.0	1.4 ± 1.8	
Moment in BW BH%	SR		0.6 ± 0.6	0.8 ± 0.8	
	CR		0.7 ± 0.6	0.8 ± 0.9	
	VC		0.7 ± 0.7	1.0 ± 1.1	

Residual forces and moment of inverse methods and tracking simulations differed considerably. For inverse methods, the mean RMS residual pelvis forces and moments over all trials were 5.9 ± 2.3 GRF max% and 0.9 ± 0.4 GRF max BH%, respectively. In contrast, marker and coordinate tracking simulations had maximum RMS residual pelvis forces of 6.0⋅10−8 GRF max%, pelvis moments of 1.7⋅10−8 GRF max BH%, and joint moments of 1.5⋅10−6 GRFmax BH%.

The scaled models, the experimental data, the result of the inverse methods, and the simulation results are provided online (Nitschke et al., 2022).

Discussion

In this article, we showed that it is feasible to reconstruct measured motions by directly tracking marker and GRF data without task constraint in an optimal control simulation of a 3D full-body musculoskeletal model. We successfully tracked COD running motions without prior knowledge of the task nor the initial state. The presented formulation of the dynamics is therefore suited to reconstruct arbitrary motions. Marker tracking was superior to coordinate tracking and comparable to inverse methods in terms of marker errors while resulting in mutually and dynamically consistent kinematics and kinetics.

Marker tracking simulations took approximately three to four times longer to solve than coordinate tracking simulations (see Table 2) and therefore were computationally much more expensive. At the same time, marker tracking did not require considerably more CPU time per iteration, implying that the higher number of iterations mainly caused the large increase in computation time. The marker tracking has a higher complexity due to a higher non-linearity in the objective and gradients since marker positions must be obtained from the model states. In contrast, the generalized coordinates are part of the states and, therefore, optimization variables. Consequently, alternative optimization algorithms or formulations of the problem would have to be investigated to decrease the number of iterations and thus the computation time of marker tracking. In conjunction with this, the ratio of CPU time spent in the function evaluation of the NLP was higher for marker tracking since marker positions must be computed in every iteration.

Despite the recent advances, solving optimal control problems is computationally demanding, while inverse methods can be computed in seconds or minutes. However, the time required to solve an optimal control simulation highly depends on the formulation of the problem and its implementation. In this work, we used a large number of collocation nodes of up to 180 by reconstructing the motion at the original sampling frequency of 175 Hz, resulting in a large number of optimization variables and thus unknowns of up to about 80,000. In comparison, Venne et al. (2022) generated simulations with up to 106 nodes and 12,444 variables using multiple shooting. Furthermore, the choice of initial guess affects the time required for the optimization. We initiated the optimal control problem using a standing simulation to have an unbiased initial guess. Instead, the solution from inverse methods could be used, which decreases the time required since this initial guess is closer to the final solution. We chose not to do this because this would bias the simulations towards the result of the inverse methods and not allow for an independent comparison between the three methods. Therefore, optimization could be further accelerated by reducing the number of collocation nodes and using an informed initial guess depending on the application’s specific requirements.

Measured marker positions were tracked closest by inverse kinematics (see Table 3) since it estimates the kinematics for each time point separately without accounting for model dynamics. Neglecting dynamics makes inverse kinematics prone to track measurement noise and soft tissue artifacts, leading to smaller marker errors but also to high-frequency components in the movement (see Fig. 3B). Those high-frequency components make it necessary to filter estimated coordinates before using them as input for inverse dynamics. However, the choice of the cut-off frequency can considerably influence the resulting joint moments (Derrick et al., 2020). In contrast to inverse methods, optimal control simulation acts as a physical filter by accounting for model dynamics and minimizing effort in the objective. This eliminates the need to filter the data in advance but requires to balance tracking and effort in the objective to find a trade-off between close tracking and realistic neural excitation patterns.

Inverse kinematics tracked markers at the feet more closely than the simulations, even though marker positions were strongly affected by shoe deformation. The close tracking of the deformed and thus inaccurate marker positions resulted in unrealistically high plantarflexion of the mtp joint for inverse kinematics (see Fig. 3B). Similar to soft tissue artifacts, the deformations of the shoes are not modeled by the musculoskeletal model since virtual markers are rigidly attached to the model. However, in the optimal control simulations, high plantarflexion was prevented by modeling passive moments for all joints which became active in the mtp joint when plantarflexion exceeded 8 degrees (see supplemental information of Nitschke et al. (2020)). Consequently, the feet marker are then not tracked strictly, which results in higher marker errors for the simulation. In inverse kinematics and marker tracking simulation, the mtp joint angle could become more realistic by weighting the markers on the deformed shoe less than the other markers. In any case, weights should be adjusted only if it is appropriate considering the data, application, and biomechanical variables of interest since it might worsen the estimation of other variables and requires hand-tuning.

For marker and coordinate tracking simulation, input variables were tracked more closely than the variables not used in the objective of the optimization (see Table 3). Coordinate tracking follows the recorded marker data worst since errors propagate by tracking inaccuracies in the coordinates estimated with inverse kinematics. In detail, inaccuracies resulting from measurement, inverse kinematics, and coordinate tracking add up in contrast to marker tracking, where only the inaccuracies resulting from the measurement and tracking add up. Furthermore, errors made in the tracking of individual joint angles accumulate along the kinematic chain and result in a larger difference in the position of distal segments and thus of the marker positions. Therefore, distal segments like the hands and feet deviate considerably for coordinate tracking from the other two reconstruction methods (see Fig. 3A). These findings are in agreement with previous work for skeletal models (Febrer-Nafría et al., 2020; Venne et al., 2022) and proof that marker tracking is more accurate than coordinate tracking in terms of marker error. However, inverse kinematics which we used for comparison is not a gold standard since it is not reflecting the bone motion, but is subject to errors. Hence, estimated kinematics should be evaluated with bone pins or medical imaging. Regardless of the reconstruction method chosen, we strongly recommend analyzing the marker error carefully since this is the only available error measure when performing optical motion capturing.

Marker tracking reconstructed the measured GRFs slightly better than coordinate tracking (see Table 3). Even though we used the same weight of the GRF tracking term in both types of simulations (see Table 1), it might be that the GRF term had a higher influence on the overall objective in marker tracking than in coordinate tracking as the kinematic tracking data differed. Therefore, adjusting the weighting in the objective could counteract the different accuracies with respect to the GRFs, but would change the relation between GRF tracking and effort term.

The difference in GRF tracking between the marker and coordinate tracking might have also caused the slight difference in the reconstruction of joint moments. Again, it is necessary to note that the reference joint moments obtained with inverse dynamics do not represent a gold standard. However, joint moments can only be measured using instrumented implants. Alternatively to the comparison with inverse methods, the simulation could be evaluated by reconstructing simulated motions. Simulated data has the advantage that the ground truth would be known. Nevertheless, an evaluation with simulated data could hardly reflect all characteristics of real-world data like noise, errors, and soft tissue artifacts perfectly.

Although the constraint we introduced to remove the need for a task constraint (see Eq. (8)) allows the reconstruction of various motions, one minor downside is that it requires additional samples before the motion of interest. However, we also recommend analyzing longer time periods when applying inverse methods to reduce filtering artifacts. When reconstructing longer motions with simulation, it could be investigated to use moving horizon estimation (Bailly et al., 2021) where a new simulation is initiated using the end of the last simulation.

The analysis of residual forces and moments for inverse methods and tracking simulation highlights the difference between the two methods with respect to dynamic inconsistencies. In this study, inverse methods led to slightly higher residuals than recommended by Hicks et al. (2015). They recommend RMS residual forces lower than 5 GRF max% and residual moments lower than 1% of GRF max times center of mass (CoM). Residuals in inverse methods could be reduced by manually adjusting the inertial parameters of the scaled model (Hicks et al., 2015). However, a manual adjustment to every participant is hardly feasible for large studies or in automated analysis pipelines. In contrast to inverse methods, the optimal control simulations had negligibly small residuals. As a result, estimated biomechanical variables are dynamically consistent. Therefore, there are no inconsistencies between the cause, i.e., neural excitation of the muscles, and the effect, i.e., the resulting motion. However, it has to be kept in mind that the simulation, in return, assumes that the dynamic model is perfect.

Potential inaccuracies and simplifications in musculoskeletal models can limit all reconstruction methods, i.e., inverse methods and optimal control simulation. In this study, we scaled the musculoskeletal model using marker positions of a static trial but did not personalize any muscle properties. For both simulation methods, it might be possible to further reduce tracking errors and improve muscle variable estimation when a better estimate of muscle parameters is available, for example, from strength tests (Hegarty et al., 2019) or medical imaging (Valente et al., 2017).

In the future, tracking marker positions directly instead of estimated generalized coordinates could offer further possibilities. It would, for example, allow personalizing model parameters based on measurement data within the optimal control problem by adding certain parameters (e.g., segment lengths) to the optimization variables instead of predefining them. Furthermore, simulations can be created even when marker data is incomplete, for example, due to occlusions (Venne et al., 2022). The time periods where marker data is missing can be excluded from the tracking objective since model dynamics and effort minimization will still produce a realistic movement for these periods. But most importantly, marker tracking simulations could also be driven by virtual marker positions extracted from video data, depth images, or radar technology instead of using marker-based optical motion capturing.

Conclusions

In conclusion, we proved that it is feasible to directly drive optimal control simulations by marker and GRF data for 3D full-body musculoskeletal models to reconstruct COD running motions without estimating generalized coordinates in an intermediate step. We presented a detailed comparison of marker tracking simulations, coordinate tracking simulations, and inverse methods. In contrast to inverse kinematics and dynamics, optimal control simulation returns kinematics and kinetics, which are mutually and dynamically consistent. Dynamic consistency is especially important for the analysis of fast motions for example in sport science. Our results confirmed that marker tracking reconstructs measured marker positions more accurately than coordinate tracking. We, therefore, recommend using marker tracking simulations over coordinate tracking for reconstructive simulations, especially for applications investigating small changes in kinematics or kinetics. Nevertheless, coordinate tracking might still be advantageous when reference data is included in predictive simulations.

Supplemental Information

Supplemental Information 1 Frontal view of kinematics of a v-cut.

The result of inverse kinematics, coordinate tracking, and marker tracking is represented in red, orange, and green, respectively. The measured marker positions are displayed in blue. The motion had in total 149 samples at 175 Hz.

Click here for additional data file.

Supplemental Information 2 Lateral view of kinematics of a v-cut.

The result of inverse kinematics, coordinate tracking, and marker tracking is represented in red, orange, and green, respectively. The measured marker positions are displayed in blue. The motion had in total 149 samples at 175 Hz.

Click here for additional data file.

Additional Information and Declarations

Competing Interests

Author Contributions

Human Ethics

Data Availability

The authors declare that they have no competing interests.

Marlies Nitschke conceived and designed the experiments, performed the experiments, analyzed the data, prepared figures and/or tables, authored or reviewed drafts of the article, developed and implemented the algorithms, and approved the final draft.

Robert Marzilger conceived and designed the experiments, authored or reviewed drafts of the article, and approved the final draft.

Sigrid Leyendecker conceived and designed the experiments, authored or reviewed drafts of the article, and approved the final draft.

Bjoern M. Eskofier conceived and designed the experiments, authored or reviewed drafts of the article, and approved the final draft.

Anne D. Koelewijn conceived and designed the experiments, authored or reviewed drafts of the article, and approved the final draft.

The following information was supplied relating to ethical approvals (i.e., approving body and any reference numbers):

The ethics committee of the Friedrich-Alexander-Universität Erlangen-Nürnberg (Re.-No. 106_13 B) approved the study, and participants gave informed consent before participation.

The following information was supplied regarding data availability:

The data is available at Zenodo: Nitschke, Marlies, Marzilger, Robert, Leyendecker, Sigrid, Eskofier, Bjoern M., & Koelewijn, Anne D. (2022). Optical motion capturing of change of direction motions reconstructed with inverse kinematics and dynamics and optimal control simulation [Data set]. Zenodo. https://doi.org/10.5281/zenodo.6949012.

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
