# Peer review of "Change the direction: 3D optimal control simulation by directly tracking marker and ground reaction force data"

_PeerJ, doi:10.7717/peerj.14852_

## Round 0.1 · original submission · Major Revisions

The reviewers were generally impressed by the fairly novel technical element of the study in performing this variety of data-tracking with a complex model. There were some concerns on the clarity and motivation of the methodological choices and acknowledgement of previous related work that should be addressed.

·

Basic reporting

Please see 'Additional comments' for my full review

Experimental design

Please see 'Additional comments' for my full review

Validity of the findings

Please see 'Additional comments' for my full review

Additional comments

Summary:

The authors have presented a comparison between 3 different reconstruction methods: inverse methods (inverse kinematics + inverse dynamics), optimal control problem (OCP) that tracks the generalized coordinates (from IK), and an OCP that tracks the markers. The comparison was made to 3 trials of straight running, curved running, and a v-cut trial across 10 participants. Inverse kinematics yielded the most accurate marker tracking, with a close second from the marker tracking OCP, and the worst performance from the generalized coordinate tracking.

This is a very useful paper to have in the literature. I hope that the methods available improve so that more people are able to take advantage of OCP reconstruction methods.

About your reviewer:

I've applied optimal control methods to musculoskeletal models in my own research for the past five years, have authored several muscle models, several foot-ground contact models. Although I have mostly used direct multiple shooting, rather than direct collocation, the methods and numerical issues are similar.

Abstract

Introduction

1. Somewhere in the paragraph that begins on line 88 there should be a reference to bioptim: because its open-source and supports both direct multiple shooting and direct collocation. This sentence would be a good spot for it:

"Furthermore, the increasing availability of toolboxes facilitates access to the methodology (Dembia et al., 2020; Patterson and Rao, 2014)."

Michaud B, Bailly F, Charbonneau E, Ceglia A, Sanchez L, Begon M. Bioptim, a python framework for musculoskeletal optimal control in biomechanics. IEEE Transactions on Systems, Man, and Cybernetics: Systems. 2022 Jun 28.

https://github.com/pyomeca/bioptim

2. Lin et al. is related work that should be mentioned somewhere in the literature review

Lin YC, Walter JP, Pandy MG. Predictive simulations of neuromuscular coordination and joint-contact loading in human gait. Annals of biomedical engineering. 2018 Aug;46(8):1216-27.

3. line 114: add a pair of commas around "for example"

4. line 118: the paragraph that begins "Furthermore, evaluation" belongs with the previous paragraph. The lines from 118-122 are punctuated as a paragraph, but this is not a paragraph: it does not meet the minimum requirements of having a topic sentence, a body sentence, and a sentence linking to the next paragraph.

Methods:

5. line 150: add a pair of commas around "but did not extract"

6. Eqns. 1-5: This is optional, as it is a matter of taste: The very long subscripts (e.g. N_{states}, N_{controls}) are a bit awkward to read. Consider just shortening the subscripts to a single character, for example N_{X} for N_{states}, N_{U} for N_{controls}, N_{\tau} for N_{tor}, etc. Most of these single variables have been defined already, so using them as subscripts is quite clear.

7. line 158-160

It's unclear to me what is meant by:

"In contrast to other musculoskeletal models, runMaD has an adapted pelvis rotation sequence that makes the pelvis orientation interpretable independent of the movement direction."

If the pelvis defined with respect to the inertial frame using a 6 Dof free-flyer joint then the orientation of the pelvis would always be independent of the movement direction (by which I assume you mean CoM velocity). What am I missing?

8. Did the parameters of the model's muscles have to be adjusted in any way to each participant? I see from Nitschke et al. 2020 that the runMaD model makes use of Sam Hamner's musculotendon properties. Having just read the supplementary material for Nitschke et al. 2020 I didn't see any mention of the strength of the modelled muscles being adjusted. I know that Sam made a number of manual adjustments to the model he used (with parameters that originally came from a cadaver) so that it could run at 5 m/s. Was this model strong, fast, and flexible enough to complete the simulated motions without any subject specific adjustment?

9. line 257

The convergence tolerance is listed as 10^{-4}:

a. Please describe exactly how this tolerance is defined.
b. In the results, please provide the norm of the gradient of the Lagrangian so that the reader has some idea of how close the numerical solutions are to a minima.

Results

10. Optional: This is a style point, but the results text begins with a statement that the work you've done is available online. The opening sentence and first paragraph of the results section are usually reserved for the most impactful result of the study. You are doing the paper a disservice by leading with a weak statement. If this is the first time you've heard a comment like this, I suggest you give Brand & Huiskes a read and then consider restructuring the results section.

Brand RA, Huiskes R. Structural outline of an archival paper for the Journal of Biomechanics. Journal of Biomechanics. 2001 Nov 1;34(11):1371-4.

11. Please add a plot or table comparing the residual forces of a typical IK + ID approach to the residual forces that are present in each of the simulations. It is mentioned several times in the paper that dynamic consistency is a benefit of using optimal control for reconstruction, but the benefit is not quantified and presented in the results. I found it strange that dynamic consistency was not reported.

I was expecting that this would be the biggest result of the paper. Why? Residual forces can be huge when using traditional IK + ID, and these forces hang like a dark cloud over any kind of later analysis that depends on forces. In contrast, optimal control based reconstruction should have residual forces that are zero, or close to it, and yet offer a similar level of marker error. This is much more important than a 6-7mm improvement in average marker error. I think the paper could be much more impactful if you reported the difference in residual forces as a result.

12. Figure 3

a. Figure 3 has a wide variety of different scales for each plot which makes it challenging to interpret the plot. To make the plots easier to interpret please: first, indicate in each plot the point in which the biggest pair-wise difference between the methods occurs; second, add a label that denotes the size of the difference. For example ToeL in Z would probably have the largest disagreement at frame 15, and the biggest pair-wise difference would be roughly 60 mm.

b. The last row of plots have y-axis labels of normalized moment but the titles of angle.

c. Please add plots comparing the center-of-pressure location between the measurements and the models. I realize that technically this information is present in ankle, subtalar, and mtp joint moment plots. However, it is a lot easier to interpret a difference in center-of-pressure location than it is to interpret subtalar joint moments, for example. If this doesn't fit, consider breaking up Figure 3 into a kinematics plot, and a separate kinetics plot.

13. Did any of the modelled muscles:
a. Saturate with a maximum excitation?
b. Reach a very short or very long normalized length (<0.6, or >1.5)?
c. Reach a very fast contraction speed (< -7 lopt/s or > 7 lopt/s)

It would be useful to know this information. If the answer is no to all 3 questions, then it would be nice to see a sentence somewhere in the paper stating this fact.

Discussion:

14. Line 324

"In this paper, we showed that it is feasible to reconstruct measured motions directly from marker and GRF data by optimal control simulation of a 3D full-body musculoskeletal model."

This sentence makes it sound like this paper is the first paper to present an optimal control solution in which a 3D musculoskeletal model tracks recorded data. Please qualify this statement to make it clear that other optimal control reconstructions using 3D musculoskeletal models exist (Lin et al. is comparable), but that this is the first to directly track marker data.

Lin YC, Walter JP, Pandy MG. Predictive simulations of neuromuscular coordination and joint-contact loading in human gait. Annals of biomedical engineering. 2018 Aug;46(8):1216-27.

15. Line 382 [Comment - no response required]

"Hence, estimated kinematics should be evaluated with bone markers or medical imaging."

Sadly even bone markers, if you mean bone pins, have drawbacks: skin pulls on the pin and distorts the location of the marker.

16. Line 404 [Comment - no response required]

In the future, marker tracking simulations could also be driven by virtual marker positions extracted from video data, depth images, or radar technology instead of using marker-based optical motion capturing.

Having seen 77 GHz radar captures of people moving indoors, I think it will be many years before tracking can be applied solely to the output of a radar chip.

Reviewer 2 ·

Basic reporting

This manuscript is well-written and easy to read.

Experimental design

Their study rationale needs improvement.

#100-108
This reviewer suggests that including more rationale about tracking marker positions directly is more critical that tracking generalized coordinates for describing human motion using the musculoskeletal model with optimal control theory. The authors stated that the inaccuracies caused by tracking generalized coordinates in an optimization framework could be reduced by directly tracking marker position data. However, marker positions are also prone to factors including skin movement below markers. In the method (#276) and discussion (#380), the authors also mentioned this issue by stating that no ground truth can be compared with the methods used in this study. Thus, readers may be interested in hearing more about what benefit they have when tracking marker positions directly in the optimization framework.

#118-122
This reviewer suggests removing this paragraph because it does not support the study rationale.

#180
Please provide a list of tracking variables (e.g., y included #, what is j, all 42 markers were tracked).

#272
This reviewer would like to see more details about what animations the authors used to evaluate the quality of the optimal solutions. How were they generated?

# 404
This reviewer liked the idea described here. Pose estimation technique has been growing interest in the biomechanics community (e.g., OpenCap suggested by Uhlrich et al. (2022)). This reviewer thought tracking points of interest with the help of musculoskeletal modeling and optimal control could expand their application to real-world problems. Consistent with the earlier comment, the study may be better supported by including the application (tracking marker positions directly) in the Introduction.

#160-161
Please provide more details about the musculoskeletal model (e.g., the number of muscles per leg and arm, degrees of freedom for legs and arms) so that readers do not need to find the model paper. This reviewer does not expect too many details for that model though.

Validity of the findings

Their finding may be impacted by different weighting values, which it needs to be addressed as a study limitation.

#169, #313, #366, #388
Would authors provide their thoughts on the effects of unequal weighting values for markers and tracking variables on their findings? For example, the authors stated that all markers were weighted equally in inverse kinematics. The observed discrepancies, especially for foot-related makers and angles, may be reduced or better estimated with greater weighting values for those in the inverse kinematics and objective function. If using unequal weighting values would affect their findings, please add this as a study limitation.

Additional comments

This reviewer includes comments for the manuscript in the attached document.

Reviewer 3 ·

Basic reporting

The paper is very well written and organized.

Experimental design

The experiments and simulations are appropriately designed to address the posed research question. The study is rigorous regarding the methodology applied and its description in the manuscript.

Validity of the findings

The data is made available in an organized and accessible manner, allowing for verification and use in future studies. The paper offers insightful results and discussion on the performance of the marker-tracking optimal control approach when applied to complex 3D musculoskeletal models and activities.

Additional comments

General Comments
The study investigates the feasibility and the performance of a marker-tracking optimal control approach to reconstruct kinematics and kinetics of running and change-of-direction (COD) running using a complex full-body musculoskeletal model. The analysis was performed on 10 participants (3 trials each). The authors present a detailed and careful comparison with the coordinate tracking simulation approach and the more commonly used inverse kinematics/dynamics approach.
Although marker tracking is not new, previous studies have used either models with few degrees of freedom or torque-driven models. This study investigates the performance of marker tracking for a complex, full-body musculoskeletal model, offering a straightforward and high-quality assessment of performance compared to the other two methods commonly used in the literature. The manuscript is very well written and offers many insights on the advantages and disadvantages of using a marker-based tracking optimal control approach, also regarding computational costs. Therefore, although the methodology is not new, the quality and completeness of the study and its application to a complex model and a complex activity (COD running) substantially contribute to the body of knowledge in the area. All the data is made available in a well-organized manner.

Specific Comments
- Abstract: authors mention that marker tracking is recommended when precise marker tracking is required. It would be essential to elaborate on this in the discussion session. In which instances would accurate marker tracking be required?
- Line 97 - the isolated claim that the coordinate tracking is computationally efficient because generalized coordinates are used as kinematic states is misleading. I suggest authors rephrase or displace this sentence to put it in a comparative context with the marker tracking approach.
- Lines 99 to 101 – the authors could be more precise when discussing error propagation. All methods are ultimately tracking markers. The difference is how and to which extent the marker trajectories' inconsistencies are propagated. Although the discussion is very well-written, the authors could shed more light on the problem from this perspective.
- In equations (1) to (4), I assume the denominator should be "N+N_{tra,mus,tor}" and not "N*N_{tra,mus,tor}". Please, check.
- Lines 193 to 201 - please, provide a reference or a rationale to justify the sum of volume-weighted cubed neural excitations to solve the muscle redundancy problem instead of other, more commonly used cost functions.
- Lines 208-230: perhaps incorporating initial states as additional optimization parameters could address this issue more straightforwardly. Have authors tried this? Authors don't need to perform any changes to the utilized approach, as this comment is meant to be a mere contribution to the discussion.
- Lines 328: please, mention in which sense marker tracking is comparable to inverse methods.

---

## Round 0.2 · accepted · Accept

Dr Nitschke, the original Academic Editor is no longer available and so I am making this decision in my capacity as Section Editor.

Thank you for addressing all of the concerns raised by the Reviewers for your manuscript. I am pleased to therefore recommend your amended manuscript for publication.

Thank you for supporting PeerJ and we look forward to future submissions.

Thanks, A/Prof Mike Climstein

·

Basic reporting

no comment

Experimental design

Overall I'm very pleased with the work that the authors have done to address my comments. There are 3 further comments which the authors should consider addressing in the paper.

1. In the response to 1.8 and 1.13 it is mentioned:

"Although the scaling of activation is equivalent to the scaling of maximum isometric force, we are aware that scaling the activation is less common."

Solely scaling the activation is not equivalent to scaling the maximum isometric force: the force-length curves of the tendon and the parallel element are usually constructed so that they scale with the maximum isometric force, but do not scale with activation. By only scaling activation you may have created an MTU that has a CE that can generate 5x the active force, but has a tendon and parallel element that has (1/5) the stiffness that would be expected. For example, applying activation scaling to a typical soleus muscle and Achilles tendon with an isometric strain of 4.9% [1] would result in a soleus muscle with 5x the strength and an Achilles tendon with an isometric strain of around 24.5%. This is extreme: the Achilles tendon has gone from a typical value [1] to being larger than any in-vivo measurements of which I'm aware [2].

Were the force-length curves of the tendon and the parallel element also scaled by 5? Or are the simulated tendons very compliant?

[1] Magnusson, S. P., Aagaard, P., Dyhre-Poulsen, P. & Kjaer, M. Load-displacement properties of the human triceps surae aponeurosis in vivo. The Journal of Physiology 531, 277–288 (2001).

[2] Waugh, C. M., Blazevich, A. J., Fath, F. & Korff, T. Age-related changes in mechanical properties of the Achilles tendon. Journal of anatomy 220, 144–155 (2012).

2. [Optional] In the response to Comment 1.13 and 1.8 the authors helpfully mentioned the range of normalized CE lengths and CE velocities that occurred during the simulations. However, these ranges were not mentioned in the paper. It would be nice to see one more sentence that notes:

"Some of the simulations had wide range of active fiber lengths (0.24 to 1.59 lopt) and velocities (-15.76 to 9.25 lopt/sec) like to due to the speed of the movement and small differences between the model and the participant."

These differences are not uncommon and the field would benefit if there were more publications that were transparent that these kinds of differences can exist when a scaled generic model is used to follow rapid movements.

3. [Optional] In response to comment 1.12 the authors presented CoP profiles but have decided not to include these plots because the profiles can differ between reconstruction methods. This is very interesting and valuable information, and it is a great reason to include these plots in the paper. The COP is a valuable output that many people reading this paper will care out. It is ultimately your choice, but I think the paper is more valuable if the CoP data is included, and the differences between the CoP profiles is addressed in the discussion somewhere.

Validity of the findings

no comment

Additional comments

no comment

Reviewer 2 ·

Basic reporting

None

Experimental design

None

Validity of the findings

None

Additional comments

The authors responded to the comments raised by this reviewer well. This reviewer did not have any further comments and appreciated all the sincere responses.

Reviewer 3 ·

Basic reporting

The paper is very well written and organized.

Experimental design

The experiments and simulations are appropriately designed to address the posed research question. The study is rigorous regarding the methodology applied and its description in the manuscript.

Validity of the findings

The data is made available in an organized and accessible manner, allowing for verification and use in future studies. The paper offers insightful results and discussion on the performance of the marker-tracking optimal control approach when applied to complex 3D musculoskeletal models and activities.

Additional comments

My comments and concerns were fully and properly addressed by the authors.

---

## Author Rebuttal · Round 0.2

**Friedrich-Alexander-Universität**
**Department Artificial Intelligence**
**in Biomedical Engineering | AIBE**

Machine Learning and Data Analytics Lab (Department AIBE)

Ross Miller
*Academic Editor, PeerJ*

**Department**
**Artificial Intelligence in**
**Biomedical Engineering (AIBE)**

**Machine Learning and**
**Data Analytics Lab**

Marlies Nitschke, M.Sc.

Carl-Thiersch-Straße 2b
91052 Erlangen
+49 9131 85-27 890

Marlies.nitschke@fau.de
www.mad.tf.fau.de

Erlangen, 23. Oktober 2022

## Responses to the reviewer's comments

Dear Dr. Miller, dear reviewers,

We thank you and the reviewers for your effort in evaluating our manuscript, "Change the direction: 3D optimal control simulation by directly tracking marker and ground reaction force data".

We highly appreciate the valuable comments. We agree with most of the reviewers' points and revised the manuscript carefully. We rewrote relevant passages to improve the clarity of the presentation. We highlighted all changes in the manuscript and listed our responses to individual comments in the following:

We hope we can eliminate the concerns with the revised manuscript and believe that the manuscript is now ready for publication.

Marlies Nitschke
*(On behalf of all authors)*

# Responses to reviewer #1 (Matthew Millard)

## Summary

The authors have presented a comparison between 3 different reconstruction methods: inverse methods (inverse kinematics + inverse dynamics), optimal control problem (OCP) that tracks the generalized coordinates (from IK), and an OCP that tracks the markers. The comparison was made to 3 trials of straight running, curved running, and a v-cut trial across 10 participants. Inverse kinematics yielded the most accurate marker tracking, with a close second from the marker tracking OCP, and the worst performance from the generalized coordinate tracking.

This is a very useful paper to have in the literature. I hope that the methods available improve so that more people are able to take advantage of OCP reconstruction methods.

## Response

We are really pleased that the reviewer underscored the relevance of the paper to the community. We thank the reviewer for the valuable comments which helped us to improve the manuscript.

## Comment 1.1

Somewhere in the paragraph that begins on line 88 there should be a reference to bioptim: because its open-source and supports both direct multiple shooting and direct collocation. This sentence would be a good spot for it: "Furthermore, the increasing availability of toolboxes facilitates access to the methodology (Dembia et al., 2020; Patterson and Rao, 2014)."
Michaud B, Bailly F, Charbonneau E, Ceglia A, Sanchez L, Begon M. Bioptim, a python framework for musculoskeletal optimal control in biomechanics. IEEE Transactions on Systems, Man, and Cybernetics: Systems. 2022 Jun 28.
https://github.com/pyomeca/bioptim

## Response 1.1

We regret that we missed citing this important contribution even though we cited other work of the group. Bioptim is a great example of open-source code and should there be credited with the recognition it deserves. Therefore, we added the reference to the proposed sentence:
*"Furthermore, the increasing availability of toolboxes facilitates access to the methodology (Dembia et al., 2020; Michaud et al., 2022; Patterson and Rao, 2014)."*

## Comment 1.2

Lin et al. is related work that should be mentioned somewhere in the literature review.
Lin YC, Walter JP, Pandy MG. Predictive simulations of neuromuscular coordination and joint-contact loading in human gait. Annals of biomedical engineering. 2018 Aug;46(8):1216-27.

## Response 1.2

We agree with the reviewer that the work of Lin et al. should be included in the literature review. Since we are mostly focusing on reconstructive simulation and not on predictive simulations, we added the reference in the following two sentences:

- *"Optimal control simulations cannot only be used to reconstruct measured motions but also to predict responses to environmental changes (Dorschky et al., 2019; van den Bogert et al., 2011, 2012) or task changes (Lin et al., 2018; Nitschke et al., 2020). "*

[Figure]

- *"Reconstructive optimal control simulations are traditionally generated by tracking generalized coordinates or joint angles in combination with measured GRFs (e.g., Dembia et al. (2020); Haralabidis et al. (2021); Heinrich et al. (2014); Lin et al. (2018); Nitschke et al. (2020); van den Bogert et al. (2011,2012); see Fig. 1b)."*

## Comment 1.3
line 114: add a pair of commas around "for example"

## Response 1.3
We have added a pair of commas.

## Comment 1.4
line 118: the paragraph that begins "Furthermore, evaluation" belongs with the previous paragraph. The lines from 118-122 are punctuated as a paragraph, but this is not a paragraph: it does not meet the minimum requirements of having a topic sentence, a body sentence, and a sentence linking to the next paragraph.

## Response 1.4
We agree with the reviewer that the two paragraphs belong to each other and that the last two sentences, beginning with "Furthermore, evaluation", do not qualify as a standalone paragraph. Hence, we adapted the formatting to combine the two paragraphs and adapted the first two sentences of the new paragraph as follows to better reflect the topic:

*"Overall, previous research on marker tracking was limited to small models with few optimization variables and was limited to an evaluation with simulated or little data. The models previously used for marker tracking were either musculoskeletal models with only a few DoFs (Bailly et al., 2021; Bélaise et al., 2018a,b; Moissenet et al., 2019) or skeletal models (Febrer-Nafría et al., 2020; Hoffmann et al., 2020; Venne et al., 2022)."*

## Comment 1.5
line 150: add a pair of commas around "but did not extract"

## Response 1.5
We have added a pair of commas.

## Comment 1.6
Eqns. 1-5: This is optional, as it is a matter of taste: The very long subscripts (e.g. $N_{states}$, $N_{controls}$) are a bit awkward to read. Consider just shortening the subscripts to a single character, for example $N_{X}$ for $N_{states}$, $N_{U}$ for $N_{controls}$, $N_{\tau}$ for $N_{tor}$, etc. Most of these single variables have been defined already, so using them as subscripts is quite clear.

## Response 1.6
We agree with the reviewer that a variable as a subscript might be easier to access for readers which are used the variable names. However, we think that more descriptive, but therefore longer, subscripts are easier to access for a wider readership that might be less experienced. Therefore, we decided to keep the subscripts as they were.

## Comment 1.7

line 158-160: It's unclear to me what is meant by: "In contrast to other musculoskeletal models, runMaD has an adapted pelvis rotation sequence that makes the pelvis orientation interpretable independent of the movement direction."
If the pelvis defined with respect to the inertial frame using a 6 Dof free-flyer joint then the orientation of the pelvis would always be independent of the movement direction (by which I assume you mean CoM velocity). What am I missing?

## Response 1.7

We regret that the description was not clear. The translation and orientation of the pelvis were defined with a 6 DoF free joint with respect to the global coordinate system. The original model of Sam Hamner has the following sequence/order of pelvis rotations with respect to the global coordinate system: tilt around the global z-axis, list around the sagittal body axis, and rotation around the longitudinal body axis. This means that the tilt is applied first resulting in a change of the sagittal and longitudinal body axes. Hence, if the model tilts the pelvis by 10 degrees and is not running along the global x-axis but along the global z-axis, i.e., has a pelvis rotation of 90 degrees, the model leans to the side (see Figure 1 (left) in this document). However, clinically pelvis tilt is interpreted as anterior-posterior motion.

Our adapted version runMaD has the following sequence of pelvis rotations with respect to the global coordinate system: rotation around the global y-axis, obliquity around the sagitally body axis, and tilt around frontal body axis. Hence, in this case, the rotation around the global y-axis is applied first and, therefore, obliquity and tilt are independent of the motion direction. When considering the same example (90 degrees rotation, 10 degrees tilt), the model tilts the pelvis in the anterior-posterior direction (see Figure 1 (right) in this document). Therefore, pelvis obliquity and tilt can be interpreted without considering the movement direction and, therefore, the pelvis rotation around the global vertical axis (y-axis). The following paper gives further information: Baker, R. Pelvic angles: a mathematically rigorous definition which is consistent with a conventional clinical understanding of the terms. Gait & Posture 13, 1–6 (2001).

We adapted the sentence as follows to be more precise in our explanation while trying not to distract from the main methodology of the paper:

*"In contrast to other musculoskeletal models, runMaD has an adapted pelvis rotation sequence that makes the pelvis obliquity and tilt interpretable according to their clinical definition independent of the movement direction, i.e., independent of the rotation around the vertical axis (Baker, 2001)."*

[Figure]

*Figure 1: Original model (left) and adapted model (right) with 90 degrees pelvis rotation and 10 degrees pelvis tilt.*

### Comment 1.8

Did the parameters of the model's muscles have to be adjusted in any way to each participant? I see from Nitschke et al. 2020 that the runMaD model makes use of Sam Hamner's musculotendon properties. Having just read the supplementary material for Nitschke et al. 2020 I didn't see any mention of the strength of the modelled muscles being adjusted. I know that Sam made a number of manual adjustments to the model he used (with parameters that originally came from a cadaver) so that it could run at 5 m/s. Was this model strong, fast, and flexible enough to complete the simulated motions without any subject specific adjustment?

### Comment 1.13

Did any of the modelled muscles:
a. Saturate with a maximum excitation?
b. Reach a very short or very long normalized length (<0.6, or >1.5)?
c. Reach a very fast contraction speed (< -7 lopt/s or > 7 lopt/s)
It would be useful to know this information. If the answer is no to all 3 questions, then it would be nice to see a sentence somewhere in the paper stating this fact.

### Response 1.8 and 1.13

We thank the reviewer for his interest and this detailed question. Due to the similarity of comment 1.8 with comment 1.13, both comments will be answered in this response.

The parameters of the muscles of runMaD correspond to those of Sam Hamner's published model and have not been adjusted or scaled for Nitschke et al. 2020 or for this publication. However, we allowed a maximum activation and excitation of 5 in the simulation (see table S2 in the supplementary information of Nitschke et al. 2020) to account for weak muscles. Although the scaling of the activation is equivalent to the scaling of the maximum isometric force, we are aware that scaling the activation is less common.

Based on the reviewer's comment, we checked the range of the muscle excitation, normalized length, and contraction speed for all 92 muscle-tendon-units of the 180 simulations again. In the following are the answers to the questions in comment 1.13:

* 1.13 a): The excitation of the right flexor digitorum saturates for one v-cut simulation, and the excitation of the right flexor hallucis saturates for two v-cut simulations of the same participant.

[Figure]

- 1.13 b): The normalized length reaches values from 0.24 lopt to 1.59 lopt when taking all muscle-tendon-units and simulations into account. The normalized length was smaller than 0.6 lopt for 54 (e.g., right and left gluteus medius 1) of the 92 muscle-tendon-units in at least one simulation. The normalized length was only greater than 1.5 lopt for right and left gemellus at least in one simulation.
- 1.13 c): The normalized contraction speed reached values from -15.76 lopt/s to 9.24 lopt/s when taking all muscle-tendon-units and simulations into account. The normalized contraction speed was smaller than -10 lopt/s, which was used as the maximum shortening velocity according to Thelen (see equation S7 in the supplementary information of Nitschke et al. 2020), for 16 muscle-tendon-units in at least one simulation. The normalized contraction speed was greater than 7 lopt/s for the right piriformis, soleus, and internal abdominal oblique at least in one simulation.

The ranges of motion were adapted as described in the supplementary information of Nitschke et al. 2020:

*"Pronation and supination of the foot was enabled by unlocking the subtalar joint. The metatarsophalangeal (mtp) joint was also unlocked for roll over of the foot. For both joints, a range of motion from -90° to 90° was allowed. In order to fit the recorded data of fast running, the upper limit of knee flexion was increased from 120° to 160°. Additionally, the range of motion of the pronation/supination angle at the elbow was enlarged from [0°;90°] to [0°;150°] and the default pronation/supination angle was set to 90° such that the palms were pointing towards the body for zero torque. The default elbow flexion was set to 5° to be within the range of motion."*

Based on the reviewer's comment, we checked whether the ranges of motion of the model were suitable for the 180 performed simulations again. All ranges were suitable for the reconstructed motions. However, larger ranges of motion of the arms might be needed when more dynamic motions with larger compensating arm movements are reconstructed.

As the model development and personalization were not the objectives of this paper, we decided not to elaborate on the details of the muscle properties within the paper. However, in the methods, we added the following to clarify that muscle properties were not adapted:

*"We scaled the model for every participant using the static trial in N-pose in OpenSim 4.3 (Seth et al., 2018) but did not personalize any muscle properties."*

Additionally, we added the following paragraph in the discussion:

*"Potential inaccuracies and simplifications in musculoskeletal models can limit all reconstruction methods, i.e., inverse methods and optimal control simulation. In this study, we scaled the musculoskeletal model using marker positions of a static trial but did not personalize any muscle properties. For both simulation methods, it might be possible to further reduce tracking errors and improve muscle variable estimation when a better estimate of muscle parameters is available, for example, from strength tests (Hegarty et al., 2019) or medical imaging (Valente et al., 2017)."*

### Comment 1.9

line 257: The convergence tolerance is listed as 10^{-4}:

a. Please describe exactly how this tolerance is defined.

b. In the results, please provide the norm of the gradient of the Lagrangian so that the reader has some idea of how close the numerical solutions are to a minima.

### Response 1.9

- a): We added the following phrase for clarification:
  *"for the scaled nonlinear program (NLP) error"*
  Further details on the definition of the NLP error can be found in equations 5 and 6 in Wächter and Biegler 2006.
- b): We thank the reviewer for pointing out that this information was missing. The NLP error reflects the gradient of the Lagrangian (see equation 5 in Wächter and Biegler 2006). Therefore, we added the scaled NLP errors to Table 2 and added the following in the first paragraph of the results:
  *"The averages of the scaled NLP errors were between $7.3·10^{-5}$ and $8.0·10^{-5}$ for straight running, curved running, and v-cut and for marker and coordinate tracking (see Table 2). The NLP errors did not differ largely depending on the motion type or tracking method."*

### Comment 1.10

Optional: This is a style point, but the results text begins with a statement that the work you've done is available online. The opening sentence and first paragraph of the results section are usually reserved for the most impactful result of the study. You are doing the paper a disservice by leading with a weak statement. If this is the first time you've heard a comment like this, I suggest you give Brand & Huiskes a read and then consider restructuring the results section.
Brand RA, Huiskes R. Structural outline of an archival paper for the Journal of Biomechanics. Journal of Biomechanics. 2001 Nov 1;34(11):1371-4.

### Response 1.10

We thank the reviewer for pointing this out and providing further information. Due to the emphasis of PeerJ on open science, we started the result section with the data availability. However, we also agree with the reviewer that the section should not be started with a weak statement. Therefore, we moved the sentence to the end of the result section.

### Comment 1.11

Please add a plot or table comparing the residual forces of a typical IK + ID approach to the residual forces that are present in each of the simulations. It is mentioned several times in the paper that dynamic consistency is a benefit of using optimal control for reconstruction, but the benefit is not quantified and presented in the results. I found it strange that dynamic consistency was not reported.
I was expecting that this would be the biggest result of the paper. Why? Residual forces can be huge when using traditional IK + ID, and these forces hang like a dark cloud over any kind of later analysis that depends on forces. In contrast, optimal control based reconstruction should have residual forces that are zero, or close to it, and yet offer a similar level of marker error. This is much more important than a 6-7mm improvement in average marker error. I think the paper could be much more impactful if you reported the difference in residual forces as a result.

### Response 1.11

We thank the reviewer for bringing this point up. We initially did not report the residuals since the residuals of the optimal control simulation are zero by definition except for the constraint violations.

However, we agree with the reviewer that reporting numbers for the residuals especially for ID would further strengthen the paper. Therefore, we added a paragraph in the evaluation subsection of the methods, in the results, and in the discussion. As a result of those changes in the discussion, we have further restructured the last paragraph in the discussion to also put more emphasis on the tracking of virtual marker positions from alternative technologies (see response 2.5).

## Comment 1.12
Figure 3

a. Figure 3 has a wide variety of different scales for each plot which makes it challenging to interpret the plot. To make the plots easier to interpret please: first, indicate in each plot the point in which the biggest pair-wise difference between the methods occurs; second, add a label that denotes the size of the difference. For example ToeL in Z would probably have the largest disagreement at frame 15, and the biggest pair-wise difference would be roughly 60 mm.

b. The last row of plots have y-axis labels of normalized moment but the titles of angle.

c. Please add plots comparing the center-of-pressure location between the measurements and the models. I realize that technically this information is present in ankle, subtalar, and mtp joint moment plots. However, it is a lot easier to interpret a difference in center-of-pressure location than it is to interpret subtalar joint moments, for example. If this doesn't fit, consider breaking up Figure 3 into a kinematics plot, and a separate kinetics plot.

## Response 1.12

- a): We thank the reviewer for his advice. Figure 3 indeed contains a broad variety of variables with different scales, even though we tried using only a small selection of y-axis ranges (e.g., a range of 500mm for all marker positions in the z-direction). We think that it is not advisable to add numbers since the figure reflects only one example out of 90 reconstructed motions. Some readers might tend to generalize the numbers that are given within the figure. However, based on the reviewer's comment, we investigated different annotation styles (e.g., Figure 2 in this document). We were not convinced of the additional value of the annotations. The annotations make the figures quite busy, and if the largest difference was at the first or last sample, the annotation was hardly visible for each of the annotation styles that we tried (e.g., KneL in z-direction in Figure 2 in this document where the largest error for marker tracking (red) was at the first sample). Therefore, we decided not to include an annotation in Figure 3 even though, in general, we agree with the reviewer that it might support interpretation.

[Figure]

*Figure 2: First row of Figure 3 of the paper with suggested annotation showing the point with the largest pair-wise difference to the measured marker positions for all reconstruction methods.*

- b): We thank the reviewer for pointing out the inconsistency in the figure titles. The title referred to the degrees of freedom as they are named in the model. However, we understand that this naming was confusing. Therefore, we replaced the word "angle" in all titles resulting in knee extension, ankle dorsiflexion, subtalar inversion, and mtp dorsiflexion.

- c): We agree with the reviewer that the center of pressure (CoP) might be an additional variable of interest for a detailed analysis. However, Figure 3 is supposed to only give examples of the variables evaluated in Table 3. Therefore, Figure 3 contains only a selection of marker positions and degrees of freedom of the right foot for one exemplary motion. The CoP was not tracked in this approach, nor was the RMSD evaluated for it. Furthermore, in contrast to the other variables, the CoP is only defined during the stance phase, which can differ between reconstruction methods (see Figure 3 in this document). For those reasons, we decided not to include the CoP in Figure 3.

[Figure]

*Figure 3: Part of Figure 3 with CoP over time for the respective stance phases.*

However, we agree that the interested reader should be enabled to take a more detailed look at the example. Therefore, we added the participant and trial number of the example in the results such that the provided data can be inspected in closer detail:

*"Figure 3a shows an exemplary v-cut (participant 02, trial 130 in Nitschke et al. (2022)), also provided as a video in the supplementary information."*

## Comment 1.13
Did any of the modelled muscles:
a. Saturate with a maximum excitation?
b. Reach a very short or very long normalized length (<0.6, or >1.5)?
c. Reach a very fast contraction speed (< -7 lopt/s or > 7 lopt/s)
It would be useful to know this information. If the answer is no to all 3 questions, then it would be nice to see a sentence somewhere in the paper stating this fact.

## Response 1.13
Please see response 1.8.

## Comment 1.14

Line 324

"In this paper, we showed that it is feasible to reconstruct measured motions directly from marker and GRF data by optimal control simulation of a 3D full-body musculoskeletal model."

This sentence makes it sound like this paper is the first paper to present an optimal control solution in which a 3D musculoskeletal model tracks recorded data. Please qualify this statement to make it clear that other optimal control reconstructions using 3D musculoskeletal models exist (Lin et al. is comparable), but that this is the first to directly track marker data.

Lin YC, Walter JP, Pandy MG. Predictive simulations of neuromuscular coordination and joint-contact loading in human gait. Annals of biomedical engineering. 2018 Aug;46(8):1216-27.

## Response 1.14

We regret that the word "directly" did not emphasize enough the tracking of marker data. Therefore, we rephrased the sentence:

*"In this paper, we showed that it is feasible to reconstruct measured motions by directly tracking marker and GRF data without task constraint in an optimal control simulation of a 3D full-body musculoskeletal model."*

## Comment 1.15

Line 382 [Comment - no response required]

"Hence, estimated kinematics should be evaluated with bone markers or medical imaging."

Sadly even bone markers, if you mean bone pins, have drawbacks: skin pulls on the pin and distorts the location of the marker.

## Response 1.15

We agree with the reviewer that also bone pins can be influenced by soft tissue. Nevertheless, bone pins are expected to be more accurate than skin markers. Even though the term "bone markers" is also used in literature, we replaced it with the more common term "bone pins".

## Comment 1.16

Line 404 [Comment - no response required]

"In the future, marker tracking simulations could also be driven by virtual marker positions extracted from video data, depth images, or radar technology instead of using marker-based optical motion capturing."

Having seen 77 GHz radar captures of people moving indoors, I think it will be many years before tracking can be applied solely to the output of a radar chip.

## Response 1.16

We thank you for sharing this insight. Radar is indeed a highly complex technology that can also offer a lot of possibilities. Radar technology does not only allow measuring the body hull but can also be used to localize beacons that are attached to the body. We are involved in the collaborative research center (SFB, Sonderforschungsbereich) EmpkinS in which radar technology will be further investigated for motion tracking.

An overview video on the project can be found on YouTube:
https://www.youtube.com/watch?v=vyXsKFoKPZ0

Further reads on the planned sensor technology can be found on the project website:
https://www.empkins.de/research/research-program/project-area-a

## Responses to reviewer #2 (Anonymous)

### Basic reporting

This manuscript is well-written and easy to read.

### Response

We thank the reviewer for this comment.

### Comment 2.1

#100-108

This reviewer suggests that including more rationale about tracking marker positions directly is more critical that tracking generalized coordinates for describing human motion using the musculoskeletal model with optimal control theory. The authors stated that the inaccuracies caused by tracking generalized coordinates in an optimization framework could be reduced by directly tracking marker position data. However, marker positions are also prone to factors including skin movement below markers. In the method (#276) and discussion (#380), the authors also mentioned this issue by stating that no ground truth can be compared with the methods used in this study. Thus, readers may be interested in hearing more about what benefit they have when tracking marker positions directly in the optimization framework.

### Response 2.1

We apologize that the rationale for tracking maker positions was not clear. We agree with the reviewer that marker positions are also prone to errors like skin movement. However, also the computation of generalized coordinates depends on the erroneous marker positions when using optical motion capture data. There are two main reasons for tracking marker positions instead of tracking estimated generalized coordinates when using optical motion capturing. First, inaccuracies that are caused by the estimation of generalized coordinates are tracked in the simulation. If errors are created in two consecutive processing steps, the errors will accumulate, which is also called error propagation. Second, errors made in the tracking of individual joint angles can accumulate along the kinematic chain, which is not the case when tracking absolute marker positions. While the first reason was already explained in the introduction, the second one was only mentioned in the discussion. We agree that both reasons should be explained already in the introduction and therefore rephrased the respective paragraph in the introduction by adding the following:

*"Additionally, individual joint angles are tracked rather than absolute positions. Therefore, tracking errors of each joint angle accumulate down the kinematic chain, which can cause larger positional differences at the end of this chain. In contrast to tracking coordinates in the simulation, tracking marker positions directly could avoid error propagation and error accumulation along the kinematic chain."*

Furthermore, we added an additional explanation to the discussion based on comment 3.3.

### Comment 2.2

#118-122

This reviewer suggests removing this paragraph because it does not support the study rationale.

### Response 2.2

We thank the reviewer for this advice. As we would like to emphasize that we generated a large number of simulations (10 participants x 3 trials x 3 motions per method) to create evidence for our conclusions, we decided to keep this paragraph in the introduction. However, we combined it with the previous paragraph to better connect it with the rest of the text and, therefore, also re-phrased the beginning of the new paragraph (see response 1.4).

### Comment 2.3

#180

Please provide a list of tracking variables (e.g., y included #, what is j, all 42 markers were tracked).

### Response 2.3

We regret the imprecise description. We added further explanation and information in the respective paragraph.

### Comment 2.4

#272

This reviewer would like to see more details about what animations the authors used to evaluate the quality of the optimal solutions. How were they generated?

### Response 2.4

We regret that this did not become clear with Figure 3 a) and the videos provided in the supplementary material. We clarified the respective sentence in the methods by stating that the visualization in OpenSim was used:

*"We visually compared the reconstructed motions of interest of inverse kinematics, coordinate tracking, and marker tracking using the visualization of the kinematics in OpenSim and trajectory graphs of marker positions, GRFs, pelvis translation, angles, and joint moments."*

### Comment 2.5

# 404

This reviewer liked the idea described here. Pose estimation technique has been growing interest in the biomechanics community (e.g., OpenCap suggested by Uhlrich et al. (2022)). This reviewer thought tracking points of interest with the help of musculoskeletal modeling and optimal control could expand their application to real-world problems. Consistent with the earlier comment, the study may be better supported by including the application (tracking marker positions directly) in the Introduction.

### Response 2.5

We thank the reviewer for the interesting perspective. We agree that the future of biomechanics will probably lie in new technologies like video cameras, depth images, or radar technology. This is a great motivation for researching marker tracking simulations. Nevertheless, we decided to keep this aspect in the discussion as future work since none of those technologies are yet evaluated in the presented paper. Based on this reviewer's first comment (comment 2.1), we already extended the reasoning for marker tracking in the introduction. Furthermore, in accordance with

the changes with respect to comment 1.11, we strengthened the referenced statement in the discussion by moving it to the end of the discussion:

*"But most importantly, marker tracking simulations could also be driven by virtual marker positions extracted from video data, depth images, or radar technology instead of using marker-based optical motion capturing. These measurement methods are promising, as they can be unobtrusive, highly accurate, and easily applicable."*

## Comment 2.6

#160-161

Please provide more details about the musculoskeletal model (e.g., the number of muscles per leg and arm, degrees of freedom for legs and arms) so that readers do not need to find the model paper. This reviewer does not expect too many details for that model though.

## Response 2.6

We thank the reviewer for pointing out that basic information about the model should directly be included in the paper. We added the information as follows:

*"The model has 33 DoFs (6 DoFs between ground and pelvis, 3 DoFs at the lumbar joint, 7 DoFs per leg, and 5 DoFs per arm), 92 MTUs in the lower body (6 MTUs at the lumbar joint, 43 MTUs per leg), and 5 torque actuators per arm."*

## Comment 2.7

Their finding may be impacted by different weighting values, which it needs to be addressed as a study limitation.

#169, #313, #366, #388

Would authors provide their thoughts on the effects of unequal weighting values for markers and tracking variables on their findings? For example, the authors stated that all markers were weighted equally in inverse kinematics. The observed discrepancies, especially for foot-related makers and angles, may be reduced or better estimated with greater weighting values for those in the inverse kinematics and objective function. If using unequal weighting values would affect their findings, please add this as a study limitation.

## Response 2.7

We thank the reviewer for the detailed observation that the weighting applied in inverse kinematics and the simulation could influence the reconstructed variables. And we regret that it seems that this was not discussed in enough detail. Since there are multiple aspects involved, we would like to divide our answer into two individual points:

- Weighting of the objective terms:
  We acknowledge that the weighting of the objective terms to each other always has certain limitations. However, we believe that we have addressed this point extensively in the methods and discussion as we explained the rationale for how we selected the weights (see methods: solution process) and discussed the necessity of weighting as follows:
  *"In contrast to inverse methods, optimal control simulation acts as a physical filter by accounting for model dynamics and minimizing effort in the objective. This eliminates the need to filter*

*the data in advance but requires to balance tracking and effort in the objective to find a trade-off between close tracking and realistic neural excitation patterns.*
*[…]*
*Even though we used the same weight of the GRF tracking term in both types of simulations (see Table 1), it might be that the GRF term had a higher influence on the overall objective in marker tracking than in coordinate tracking as the kinematic tracking data differed. Therefore, adjusting the weighting in the objective could counteract the different accuracies with respect to the GRFs, but would change the relation between GRF tracking and effort term."*

- Weighting of markers with respect to each other:
  In inverse kinematics and simulation, markers could be weighted differently with respect to each other. We agree with the reviewer that the estimation of specific angles could be improved by hand-tuning the marker weighting. The reviewer suggested using higher weights for the markers or angles on the feet to improve the estimation of foot-related variables. In our opinion, higher weights on the feet would not lead to an improvement since markers on the feet have larger artifacts and are therefore less reliable. We clarified that the deformed marker positions are less accurate by adapting the following sentence in the discussion:
  *"The close tracking of the deformed and thus inaccurate marker positions resulted in unrealistically high plantarflexion of the mtp joint for inverse kinematics (see Fig. 3b)."*
  In contrast to higher weights, lower weights on the feet could improve the estimation of foot-related variables. However, the estimation improvement of some joint angles might come at the expense of the accuracy of other joint angles especially since the ground truth is unknown. Therefore, the process of adjusting the weights is highly dependent on the data, application, and biomechanical variables of interest.
  Based on this, we decided to weight all markers equally.
  We added the following sentences to the discussion to address this point:
  *"In inverse kinematics and marker tracking simulation, the mtp joint angle could become more realistic by weighting the markers on the deformed shoe less than the other markers. In any case, weights should be adjusted only if it is appropriate considering the data, application, and biomechanical variables of interest since it might worsen the estimation of other variables and requires hand-tuning."*

## Responses to reviewer #3 (Anonymous)

### Basic reporting

The paper is very well written and organized.

### Response

We thank the reviewer for this comment.

### Experimental design

The experiments and simulations are appropriately designed to address the posed research question. The study is rigorous regarding the methodology applied and its description in the manuscript.

### Response

We thank the reviewer for this comment.

### Validity of findings

The data is made available in an organized and accessible manner, allowing for verification and use in future studies. The paper offers insightful results and discussion on the performance of the marker-tracking optimal control approach when applied to complex 3D musculoskeletal models and activities.

### Response

We thank the reviewer for this comment.

### General comments

The study investigates the feasibility and the performance of a marker-tracking optimal control approach to reconstruct kinematics and kinetics of running and change-of-direction (COD) running using a complex full-body musculoskeletal model. The analysis was performed on 10 participants (3 trials each). The authors present a detailed and careful comparison with the coordinate tracking simulation approach and the more commonly used inverse kinematics/dynamics approach.

Although marker tracking is not new, previous studies have used either models with few degrees of freedom or torque-driven models. This study investigates the performance of marker tracking for a complex, full-body musculoskeletal model, offering a straightforward and high-quality assessment of performance compared to the other two methods commonly used in the literature. The manuscript is very well written and offers many insights on the advantages and disadvantages of using a marker-based tracking optimal control approach, also regarding computational costs. Therefore, although the methodology is not new, the quality and completeness of the study and its application to a complex model and a complex activity (COD running) substantially contribute to the body of knowledge in the area. All the data is made available in a well-organized manner.

### Response

We thank the reviewer for the nice and precise summary of our work. We went through the comments carefully to further increase the quality of the manuscript.

### Comment 3.1

Abstract: authors mention that marker tracking is recommended when precise marker tracking is required. It would be essential to elaborate on this in the discussion session. In which instances would accurate marker tracking be required?

### Response 3.1

We thank the reviewer for the advice. We specified our recommendation in the conclusion as follows:

*"Our results confirmed that marker tracking reconstructs measured marker positions more accurately than coordinate tracking. We, therefore, recommend using marker tracking simulations over coordinate tracking for reconstructive simulations, especially for applications investigating small changes in kinematics or kinetics. Nevertheless, coordinate tracking might still be advantageous when reference data is included in predictive simulations."*

### Comment 3.2

Line 97 - the isolated claim that the coordinate tracking is computationally efficient because generalized coordinates are used as kinematic states is misleading. I suggest authors rephrase or displace this sentence to put it in a comparative context with the marker tracking approach.

### Response 3.2

We regret that the description was misleading. We rephrased the sentence as follows:

*"Since generalized coordinates are used as kinematic states of the model and are thus optimization variables, tracking of coordinates is computationally more efficient than tracking of other biomechanical variables which are not part of the optimization variables."*

### Comment 3.3

Lines 99 to 101 – the authors could be more precise when discussing error propagation. All methods are ultimately tracking markers. The difference is how and to which extent the marker trajectories' inconsistencies are propagated. Although the discussion is very well-written, the authors could shed more light on the problem from this perspective.

### Response 3.3

We thank the reviewer for pointing out that the description might have been insufficient. We added further explanation to the introduction and discussion. Please see response 2.1 for more details on the changes.

### Comment 3.4

In equations (1) to (4), I assume the denominator should be "N+N_{tra,mus,tor}" and not "N*N_{tra,mus,tor}". Please, check.

### Response 3.4

We thank the reviewer a lot for checking the equations in such detail. The denominator is supposed to normalize for the number of addends. For two nested sums, the total number of addends corresponds to the product of the individual numbers of addends. The following example could be considered:

Friedrich-Alexander-Universität
**Department Artificial Intelligence
in Biomedical Engineering | AIBE**

$$\sum_{i=1}^{2}\sum_{j=1}^{3} x_{ij} = x_{11} + x_{12} + x_{13} + x_{21} + x_{22} + x_{23}$$

This example has 2 times 3 equals 6 addends. Hence, the equations are correct.

### Comment 3.5

Lines 193 to 201 - please, provide a reference or a rationale to justify the sum of volume-weighted cubed neural excitations to solve the muscle redundancy problem instead of other, more commonly used cost functions.

### Response 3.5

We thank the reviewer for this comment. Using neural excitation instead of muscle activation is widely spread in the literature (e.g. van den Bogert, A. J., Blana, D., & Heinrich, D. (2011). Implicit methods for efficient musculoskeletal simulation and optimal control. Procedia IUTAM, 2(2011), 297–316). However, we agree that volume weighting is not yet common even though it was already used in literature (e.g., Ackermann, M., & van den Bogert, A. J. (2010). Optimality principles for model-based prediction of human gait. Journal of Biomechanics, 43(6), 1055–1060).

For clarity, we added the following explanation:

*"We used the muscle volume $w_{mus,i}$ of a muscle $i$ to account in the effort term for the strongly varying sizes and maximum isometric forces of the MTUs and, therefore, spread muscle recruitment more evenly (Happee and Van der Helm,1995)."*

### Comment 3.6

Lines 208-230: perhaps incorporating initial states as additional optimization parameters could address this issue more straightforwardly. Have authors tried this? Authors don't need to perform any changes to the utilized approach, as this comment is meant to be a mere contribution to the discussion.

### Response 3.6

We thank the reviewer for the interesting question but apologize that we are unsure what the reviewer means exactly by "incorporating initial states as additional optimization parameters". Prescribing the initial state as a constraint would be an alternative to our proposed methods. However, the initial state is unknown in our case and would have to be estimated, for example, with inverse kinematics. We aimed to propose an optimal control method for reconstructing arbitrary gait motions from marker and ground reaction force data without prior knowledge or application of inverse kinematics. Nevertheless, we think that it is beneficial to prescribe initial states when exploring longer motions with a moving horizon estimation. We believe that both aspects were discussed in sufficient detail in the methods (before Equation 8) and the discussion (second last paragraph).

### Comment 3.7

Lines 328: please, mention in which sense marker tracking is comparable to inverse methods.

### Response 3.7

We rephrased the sentence as follows to clarify both comparisons in this sentence were based on marker errors:

[Figure]

*"Marker tracking was superior to coordinate tracking and comparable to inverse methods in terms of marker errors while resulting in mutually and dynamically consistent kinematics and kinetics."*